# Parkinson’s Disease: The Neurodegenerative Enigma Under the “Undercurrent” of Endoplasmic Reticulum Stress

**DOI:** 10.3390/ijms26073367

**Published:** 2025-04-03

**Authors:** Xiangrui Kong, Tingting Liu, Jianshe Wei

**Affiliations:** 1Wushu College, Henan University, Kaifeng 475004, China; 20060002@vip.henu.edu.cn; 2Institute for Brain Sciences Research, School of Life Sciences, Henan University, Kaifeng 475004, China; ltt0808@henu.edu.cn

**Keywords:** Parkinson’s disease, endoplasmic reticulum stress, molecular mechanisms, therapeutic strategies, α-synuclein, unfolded protein response

## Abstract

Parkinson’s disease (PD), a prevalent neurodegenerative disorder, demonstrates the critical involvement of endoplasmic reticulum stress (ERS) in its pathogenesis. This review comprehensively examines the role and molecular mechanisms of ERS in PD. ERS represents a cellular stress response triggered by imbalances in endoplasmic reticulum (ER) homeostasis, induced by factors such as hypoxia and misfolded protein aggregation, which activate the unfolded protein response (UPR) through the inositol-requiring enzyme 1 (IRE1), protein kinase R-like endoplasmic reticulum kinase (PERK), and activating transcription factor 6 (ATF6) pathways. Clinical, animal model, and cellular studies have consistently demonstrated a strong association between PD and ERS. Abnormal expression of ERS-related molecules in PD patients’ brains and cerebrospinal fluid (CSF) correlates with disease progression. In animal models (e.g., Drosophila and mice), ERS inhibition alleviates dopaminergic neuronal damage. Cellular experiments reveal that PD-mimicking pathological conditions induce ERS, while interactions between ERS and mitochondrial dysfunction promote neuronal apoptosis. Mechanistically, (1) pathological aggregation of α-synuclein (α-syn) and ERS mutually reinforce dopaminergic neuron damage; (2) leucine-rich repeat kinase 2 (LRRK2) gene mutations induce ERS through thrombospondin-1 (THBS1)/transforming growth factor beta 1 (TGF-β1) interactions; (3) molecules such as Parkin and PTEN-induced kinase 1 (PINK1) regulate ERS in PD. Furthermore, ERS interacts with mitochondrial dysfunction, oxidative stress, and neuroinflammation to exacerbate neuronal injury. Emerging therapeutic strategies show significant potential, including artificial intelligence (AI)-assisted drug design targeting ERS pathways and precision medicine approaches exploring non-pharmacological interventions such as personalized electroacupuncture. Future research should focus on elucidating ERS-related mechanisms and identifying novel therapeutic targets to develop more effective treatments for PD patients, ultimately improving their quality of life.

## 1. Introduction

Parkinson’s disease (PD) is a common neurodegenerative disorder predominantly affecting middle-aged and elderly populations. Its hallmark clinical manifestations include resting tremor, bradykinesia, muscle rigidity, and postural instability. Pathologically, PD is characterized by the loss of dopaminergic neurons in the substantia nigra pars compacta (SNpc) and the formation of Lewy bodies (LBs), which primarily consist of abnormally aggregated α-synuclein (α-syn) [1]. The precise etiology and pathogenesis of PD remain unclear, but current evidence suggests that interactions between genetic and environmental factors drive the degeneration and death of dopaminergic neurons. Key implicated mechanisms include oxidative stress, mitochondrial dysfunction, endoplasmic reticulum stress (ERS), immune-inflammatory responses, proteolytic dysfunction, and apoptosis [2].

The endoplasmic reticulum (ER) is a vital organelle responsible for protein synthesis, folding, modification, and transport. Under external or internal stressors—such as oxidative stress, calcium imbalance, or excessive protein-folding demands—ER function becomes disrupted, leading to the accumulation of misfolded or unfolded proteins in the ER lumen, thereby triggering ERS [3]. To counteract ERS, cells activate the unfolded protein response (UPR), which restores ER homeostasis by upregulating molecular chaperones, inhibiting protein translation, and promoting the degradation of misfolded proteins [4]. However, persistent and unresolved ERS ultimately activates apoptotic pathways, resulting in cell death.

In PD, dopaminergic neurons are particularly vulnerable to ERS. These neurons possess highly complex dendritic and axonal structures, requiring extensive protein synthesis and trafficking, rendering them exquisitely sensitive to ER dysfunction [5]. Studies demonstrate significant upregulation of ERS markers—such as glucose-regulated protein 78 (GRP78), C/EBP homologous protein (CHOP), and X-box binding protein 1 (XBP-1)—in the brains of PD patients [6]. Moreover, ERS is closely linked to other pathological features of PD, including α-syn aggregation, mitochondrial dysfunction, and neuroinflammation [7]. This article aims to review the role and molecular mechanisms of ERS in PD. First, we introduce the fundamental concepts and activation pathways of ERS. Next, we explore the relationship between ERS and PD pathology. We then analyze ERS-associated molecular mechanisms in detail. Finally, we discuss the potential applications of ERS in PD diagnosis and treatment. By deepening our understanding of ERS in PD, we hope to provide novel insights for future research and therapeutic strategies.

## 2. Basic Concepts and Activation Mechanisms of ERS

### 2.1. Structure and Functions of the ER

The ER is a three-dimensional reticular membrane system in eukaryotic cells, which is composed of interconnected flattened sacs, tubules, and vesicles enclosed by a single membrane. It is widely distributed throughout the cytoplasm, with its membranes accounting for approximately 50% of the total cellular membrane area. The ER is broadly classified into two types: the rough endoplasmic reticulum (RER) and the smooth endoplasmic reticulum (SER) [8].

The RER, characterized by flattened sacs studded with ribosomes on its membrane surface, is primarily involved in the synthesis, processing, and transport of secretory proteins and membrane-bound proteins. Cells with high secretory activity—such as pancreatic cells and plasma cells—exhibit a well-developed RER, whereas undifferentiated or poorly differentiated cells (e.g., embryonic cells, tumor cells) possess a less prominent RER [8]. Proteins synthesized on the RER undergo post-translational modifications within the ER lumen, including glycosylation, hydroxylation, acylation, and disulfide bond formation. These modifications are critical for proper protein folding, stability, and functionality. For example, N-linked glycosylation, a key modification process in the RER, involves the attachment of sugar chains to nascent proteins to enhance their stability and function [9].

The SER, appearing as a network of smooth tubules and vesicles devoid of ribosomes, is a multifunctional organelle. Its morphology, abundance, intracellular distribution, and functional specialization vary significantly across cell types, developmental stages, and physiological conditions [8]. For instance, in steroidogenic cells (e.g., Leydig cells of the testes, ovarian luteal cells, adrenal cortical cells), the SER is highly developed to support steroid hormone synthesis [10]. Hepatocytes contain an abundant SER equipped with enzymes critical for detoxification, facilitating the metabolic transformation of drugs and toxins into less harmful or excretable forms [11]. In muscle cells, the SER differentiates into the sarcoplasmic reticulum, specialized for calcium ion (Ca^2^⁺) storage and release. During muscle contraction, Ca^2^⁺ is released from the sarcoplasmic reticulum, binding to troponin to trigger contraction [12].

Beyond protein processing, the ER synthesizes nearly all membrane lipids, including phospholipids and cholesterol. It also plays a central role in maintaining intracellular Ca^2^⁺ homeostasis, serving as a major Ca^2^⁺ reservoir. ER-stored Ca^2^⁺ participates in diverse cellular processes, such as signal transduction, muscle contraction, and secretion [13]. Dynamic Ca^2^⁺ balance is achieved through interactions between the ER and other organelles, including the plasma membrane and mitochondria.

### 2.2. Concept and Triggers of ERS

ERS refers to a series of cellular stress responses triggered by an imbalance in ER homeostasis under various stimuli, including hypoxia, glucose deprivation, viral infection, misfolded protein accumulation, oxidative stress, ER calcium dysregulation, abnormal glycosylation, and drug toxicity. These stressors lead to excessive accumulation of unfolded or misfolded proteins in the ER lumen, overwhelming its processing capacity [14].

Under physiological conditions, the ER maintains robust protein folding and quality control systems to ensure proper protein maturation. However, cellular stressors disrupt ER function. For example, hypoxia impairs energy supply, compromising ATP-dependent protein folding processes, resulting in unfolded protein accumulation [15]. Glucose deprivation disrupts glycosylation, increasing misfolded proteins [16]. Aging diminishes the efficiency of protein quality control systems, reducing the ER’s ability to manage folding anomalies and predisposing cells to ERS [17].

In pathological contexts, ERS is implicated in numerous diseases. In neurodegenerative disorders such as PD, mutant or abnormally expressed proteins (e.g., α-syn) misfold and aggregate, inducing ERS [18]. Other conditions, such as diabetes, cardiovascular diseases, and cancer, also involve ERS. For instance, in diabetes, hyperglycemia-induced ERS disrupts insulin synthesis and secretion [19]. In cancer, rapid proliferation and metabolic dysregulation force tumor cells to adapt via chronic ERS [20].

### 2.3. Signaling Pathways of ERS

When ERS occurs, the cell initiates the UPR to restore ER homeostasis. The UPR is mainly achieved through three signaling pathways, namely the inositol-requiring enzyme 1 (IRE1) pathway, the protein kinase R-like endoplasmic reticulum kinase (PERK) pathway, and the activating transcription factor 6 (ATF6) pathway [21].

IRE1 is a transmembrane protein with serine/threonine protein kinase activity and ribonuclease activity. Under normal conditions, IRE1 binds to the immunoglobulin-binding protein (BiP) and is in an inactive state. When ERS occurs, the number of unfolded proteins increases, and BiP binds to the unfolded proteins, thereby releasing IRE1 [22]. The activated IRE1 undergoes oligomerization and autophosphorylation, and its ribonuclease (RNase) activity is activated. It specifically splices the XBP-1 mRNA, removes an intron of 26 bases, changes the open reading frame of the XBP-1 mRNA, and translates the active XBP-1 protein [23]. The XBP-1 protein is a transcription factor. After entering the nucleus, it binds to the ERS response element (ERSE) and activates the transcription of a series of genes related to ER function, such as the molecular chaperone BiP, protein disulfide isomerase (PDI), etc. The expression products of these genes help to enhance the protein folding ability of the ER, reduce the accumulation of unfolded proteins, and restore ER homeostasis [24]. In addition, IRE1 can also participate in the regulation of apoptosis by activating the c-Jun N-terminal kinase (JNK) signaling pathway. When ERS persists and cannot be relieved, the excessive activation of the IRE1-JNK pathway can lead to apoptosis [25].

PERK is also a type I membrane protein located in the ER and belongs to the eukaryotic initiation factor 2α (eIF2α) protein kinase family. In the non-stress state, the N-terminal of PERK binds to BiP, keeping it in an inactive state. During ERS, BiP binds to the unfolded proteins and PERK is released and dimerizes and autophosphorylates to become activated. The activated PERK can specifically phosphorylate serine 51 of the eIF2α, inhibiting the activity of eIF2α [26], a key factor in the initiation of protein translation. After its phosphorylation, the overall level of protein synthesis in the cell is downregulated, thereby reducing the production of new unfolded proteins. At the same time, the PERK-eIF2α pathway can also induce the expression of activating transcription factor 4 (ATF4) [27]. After ATF4 enters the nucleus, it activates the transcription of a series of genes related to the cellular stress response, amino acid metabolism, antioxidant defense, etc., such as CHOP, growth arrest and DNA damage-inducible gene 34 (GADD34), etc. Among them, CHOP is a pro-apoptotic protein. Under continuous ERS conditions, the high expression of CHOP can induce apoptosis, while GADD34 can form a complex with protein phosphatase 1 (PP1) to dephosphorylate the phosphorylated eIF2α, restore protein synthesis, and maintain the basic survival needs of the cell [28].

ATF6 is a type II membrane protein located in the ER. Mammalian cells have two ATF6 subtypes, namely ATF6α (90 ku) and ATF6β (110 ku), and the two have similar structures. Under normal conditions, the C-terminal of ATF6 is located in the ER lumen, binds to BiP, and is in an inactive state [29]. When ERS occurs, BiP binds to the unfolded proteins, ATF6 is released, and it is transported from the ER to the Golgi apparatus. In the Golgi apparatus, ATF6 is successively cleaved by site-1 protease (S1P) and site-2 protease (S2P), releasing the N-terminal domain with a transcriptional activation function. This domain contains a basic leucine zipper (bZIP) motif. After entering the nucleus, it binds to the ERSE and activates the transcription of a series of genes related to ER biogenesis, protein folding, quality control, and degradation, such as BiP, GRP94, etc., to enhance the function of the ER and relieve ERS [30].

These three signaling pathways coordinate and interact with each other to jointly maintain ER homeostasis (Figure 1). In the early stage of the UPR, the three signaling pathways are activated simultaneously, and through different mechanisms, they reduce the accumulation of unfolded proteins and enhance the protein-folding and processing abilities of the ER [21]. If ERS persists and cannot be relieved, the UPR will initiate the apoptosis program to eliminate damaged cells and maintain the normal functions of tissues and organs.

## 3. Relationship Between ERS and PD Pathological Features

### 3.1. Clinical Evidence

Clinical studies have provided important clues about the association between PD and ERS (Figure 2). By detecting brain tissue samples of PD patients, researchers have found abnormal expressions of ERS-related molecules in the patients’ brains [31]. In brain regions such as the SN and locus coeruleus of PD patients, the expressions of ERS marker molecules such as GRP78 and CHOP are significantly upregulated. As a key sensor of ERS, the increased expression of GRP78 indicates that the ER is under stress [32]. The high expression of CHOP suggests that ERS may have induced the initiation of the apoptosis program. These results indicate that ERS does exist in the brain tissues of PD patients and may be involved in the pathological process of the disease.

Moreover, when analyzing the cerebrospinal fluid (CSF) of PD patients, changes in the levels of ERS-related proteins have also been found. Some studies have reported that the levels of GRP78 and phosphorylated eIF2α in the CSF of PD patients are significantly higher than those in the healthy control group [33]. This further supports the view that ERS is activated in PD patients. In addition, some studies have also found that the expression levels of ERS-related molecules are associated with the severity and disease course of PD patients. With the progression of the disease, the expression of ERS-related molecules in the patients’ brains gradually increases, suggesting that ERS may play an important role in the development of PD.

### 3.2. Animal Model Evidence

Studies on animal models provide a powerful means for deeply exploring the relationship between PD and ERS (Figure 2). Among numerous PD animal models, Drosophila and mouse models are widely used. In the Drosophila PD model, researchers induce Drosophila to express mutant α-syn through genetic manipulation to simulate the pathological features of human PD. As a result, these Drosophila exhibit obvious ERS responses, manifested as upregulated BiP expression, increased XBP-1 splicing, etc. [34]. BiP is one of the molecular chaperones that are earliest induced to be expressed during ERS, and its upregulated expression can help cells cope with the accumulation of unfolded proteins in the ER [35]. The splicing of XBP-1 is an important marker of the activation of the IRE1 pathway, indicating that the IRE1 pathway is activated in the Drosophila PD model [36]. Further studies have found that ERS is closely related to the neurodegeneration of Drosophila. Inhibiting ERS can reduce the damage to dopaminergic neurons in Drosophila and improve their motor function disorders. For example, reducing the expression of ERS-related genes by genetic means or treating Drosophila with ERS inhibitors can both reduce the death of dopaminergic neurons and improve the motor ability of Drosophila [37].

The mouse PD model also provides important information for the study of ERS. Commonly used mouse PD models include the 1-methyl-4-phenyl-1,2,3,6-tetrahydropyridine (MPTP)-induced model and the transgenic model. In the MPTP-induced mouse PD model, after an intraperitoneal injection of MPTP, the dopaminergic neurons in the SN of mice undergo progressive degeneration, and, at the same time, the expression of ERS-related molecules such as GRP78 and CHOP in the brain is significantly increased [38]. Studies have shown that MPTP causes abnormal energy metabolism and oxidative stress by inhibiting the activity of mitochondrial respiratory chain complex I and then triggers ERS. The activation of ERS further exacerbates the damage to dopaminergic neurons, forming a vicious cycle [39]. In the transgenic mouse model, such as mice expressing mutant α-syn, the activation of ERS and the degeneration of dopaminergic neurons have also been observed [40]. Through the study of these mouse models, the key role of ERS in the pathogenesis of PD has been revealed, providing an important basis for a deeper understanding of the pathological process of PD.

### 3.3. Cellular Experimental Evidence

Cell experiments are among the important methods for studying the relationship between PD and ERS as they can thoroughly explore the mechanism of ERS occurrence and its effects on dopaminergic neurons at the cellular level (Figure 2). In dopaminergic neuron cell lines, such as rat adrenal medulla pheochromocytoma cells and mouse midbrain dopaminergic neuron MN9D cells, cultured in vitro, the occurrence of ERS can be induced by various methods that simulate the pathological conditions of PD, such as overexpressing mutant α-syn and treating with MPP+ (1-methyl-4-phenylpyridinium ion, the active metabolite of MPTP) [41,42]. In PC12 cells overexpressing mutant α-syn, a large amount of misfolded α-syn proteins accumulate in the ER, leading to a significant upregulation of the expression of ERS-related proteins GRP78 and CHOP, accompanied by an increase in apoptosis [41]. This indicates that ERS triggered by mutant α-syn may be an important cause of apoptosis in dopaminergic neurons.

Emerging evidence supports the idea that mitochondrial dysfunction is not only a consequence of ERS, but also a primary inducer of ERS in PD. In MN9D cells treated with MPP+, mitochondrial dysfunction (e.g., impaired complex I activity and ATP depletion) directly disrupts ER homeostasis by depleting energy stores required for protein folding and Ca^2+^ buffering, thereby initiating ERS independently of upstream ER-specific stressors [42,43]. This is further supported by studies showing that pharmacological inhibition of mitochondrial reactive oxygen species (ROS) (e.g., mitoTEMPO) or restoration of ATP via exogenous supplementation attenuates ERS markers (GRP78, CHOP) in PD models [41,44]. These findings underscore that mitochondrial dysfunction can autonomously trigger ERS, suggesting therapies targeting mitochondrial health may synergize with ERS inhibitors to halt PD progression. ERS activates the UPR signaling pathways, including the IRE1, PERK, and ATF6 pathways [21]. After the IRE1 pathway is activated, XBP-1 splicing increases, promoting the expression of ER-related proteins to enhance the protein-folding ability of the ER [45]; the activation of the PERK pathway leads to the phosphorylation of eIF2α, inhibiting protein synthesis and reducing the production of new unfolded proteins [46]; the activation of the ATF6 pathway promotes the expression of ER molecular chaperones and folding enzymes, helping the ER restore homeostasis [47]. However, when ERS persists and cannot be relieved, these adaptive responses are not sufficient to maintain the normal function of the cells, ultimately leading to apoptosis [48]. Studies have also found that there is a close interaction between ERS and mitochondrial dysfunction. ERS can lead to mitochondrial Ca^2+^ overload, decreased membrane potential, and increased ROS production, further exacerbating mitochondrial dysfunction [49]; meanwhile, mitochondrial dysfunction can feed back to exacerbate ERS, forming a vicious cycle and jointly promoting the damage to dopaminergic neurons.

**Figure 2 ijms-26-03367-f002:**
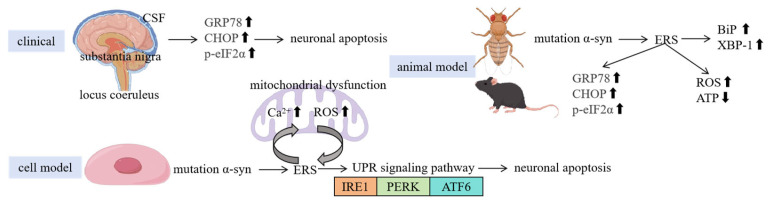
ERS is closely linked to the pathological mechanisms of PD, with clinical, animal, and cellular studies highlighting its critical role. Clinical evidence reveals significant upregulation of ERS markers (e.g., GRP78, CHOP, p-eIF2α) in PD patients’ brain regions such as the SN and locus coeruleus, as well as in CSF, correlating positively with disease severity and suggesting ERS-driven neuronal apoptosis. In animal models, Drosophila expressing mutant α-syn and MPTP-induced mouse PD models exhibit ERS activation (e.g., increased BiP expression and XBP-1 splicing), while ERS suppression alleviates dopaminergic neuron damage and improves motor function. Cellular experiments further confirm that mutant α-syn overexpression or MPP+ treatment triggers ER unfolded protein accumulation and mitochondrial dysfunction, activating the UPR pathways (IRE1, PERK, ATF6). However, persistent stress ultimately leads to apoptosis, with ERS and mitochondrial damage (Ca^2+^ overload, ROS surge) forming a vicious cycle that synergistically exacerbates neuronal degeneration. These findings collectively underscore ERS as a multifaceted contributor to PD pathogenesis and a potential therapeutic target. (↑ represents increase; ↓ represents decrease).

## 4. Molecular Mechanisms of ERS in PD

### 4.1. ERS and α-Syn

Protein α-syn is a soluble acidic protein composed of 140 amino acids, which is mainly distributed in the presynaptic terminals of the central nervous system and plays an important role in maintaining synaptic function and neurotransmitter release [50]. In PD patients, mutations or abnormal expression of the α-syn gene can lead to its misfolding and aggregation to form LBs, which are among the important pathological features of PD [51]. An increasing number of studies have shown that the abnormal aggregation of α-syn is closely related to ERS, and the two interact with each other to jointly promote the occurrence and development of PD [52]. The team led by Yu Jintai from Huashan Hospital Affiliated to Fudan University has achieved a landmark breakthrough after five years of painstaking research—for the first time globally, they discovered a new therapeutic target for PD, FAM171A2. This discovery is akin to a ray of dawn, illuminating a new journey in PD treatment research [53].

FAM171A2 is a neuronal cell membrane protein whose function has not been revealed before. Through large-scale genome-wide association analysis of the population, the team led by Yu Jintai keenly captured the close connection between FAM171A2 and PD and determined it as a PD risk gene. Located on the neuronal cell membrane, FAM171A2 plays a key role in the propagation process of pathological α-syn. In the brains of PD patients, the FAM171A2 protein content shows a significant increase, and it is positively correlated with the content of pathological α-syn in the brain [53]. FAM171A2 is considerably expressed in the vascular endothelium and microglia, which are rich in progranulin (PGRN)—a secreted pleiotropic glycoprotein associated with the development of common neurodegenerative diseases [54]. Further in vivo and in vitro experiments showed that FAM171A2 acts as a “smart key” that can selectively recognize and bind to pathological α-syn and then carry it into the neurons. Once inside the neurons, the pathological α-syn induces the misfolding of monomeric α-syn in the neurons, which, in turn, leads to neuronal death and triggers its propagation between neurons. More importantly, Wu et al. demonstrated that genetic deletion of FAM171A2 in mouse neurons significantly attenuated the progression of PD-like pathological features in transgenic models [53]. This discovery provides a new target and idea for the treatment of PD, and it is expected to intervene in PD at an early stage of the disease, block the propagation of pathological α-syn, and thus delay the progression of the disease. The research team from Renmin Hospital of Wuhan University has also made a major breakthrough in the research on the early diagnosis of PD. This study first discovered a promising new PET tracer, [18F]-F0502B, which can image the aggregated α-syn in nerve cells affected by synucleinopathies. This study has been verified in mouse models and primate models [55]. Currently, the research team is conducting clinical trials on the new PET tracer [18F]-F0502B, hoping to achieve clinical application as soon as possible.

The misfolding and aggregation of α-syn can directly trigger ERS. Under normal circumstances, α-syn exists in a soluble monomeric form and has a naturally disordered structure. When affected by factors such as gene mutations (such as A53T, A30P mutations, etc.), oxidative stress, and abnormal post-translational modifications, the structure of α-syn changes, making it prone to forming oligomers and fibrillar aggregates [56]. These abnormally aggregated α-syn can accumulate in the ER, interfering with the normal function of the ER and leading to the occurrence of ERS. Studies have found that in cell models overexpressing mutant α-syn and PD animal models, the expression of ERS-related molecules such as GRP78 and CHOP is significantly upregulated, indicating that ERS is activated. Protein α-syn aggregates can interact with molecular chaperones and folding enzymes in the ER, inhibiting their normal functions, increasing the number of unfolded or misfolded proteins in the ER, and exceeding the processing capacity of the ER, thus triggering ERS [57].

ERS, in turn, promotes the abnormal aggregation of α-syn. The activation of the UPR signaling pathway by ERS is, to a certain extent, a self-protection mechanism of the cell attempting to restore ER homeostasis [7]. However, when ERS persists and cannot be relieved, the excessive activation of the UPR signaling pathway leads to the disruption of the intracellular environment and promotes the aggregation of α-syn. For example, after the IRE1 pathway is activated, it can increase the phosphorylation level of α-syn by activating the JNK signaling pathway, and phosphorylated α-syn is more likely to aggregate [58,59]. In addition, the activation of the PERK pathway leads to the phosphorylation of eIF2α, which not only inhibits protein synthesis, but also affects the normal function of the intracellular protein quality control system, preventing α-syn from being folded and degraded correctly and in a timely manner, thus promoting its aggregation [60]. ERS can also lead to the imbalance of the intracellular redox state, generating a large amount of ROS. ROS can oxidatively modify α-syn, further promoting its aggregation [61].

The mechanism by which the interaction between the two leads to damage to dopaminergic neurons is as follows: ERS-induced apoptosis is one of the important causes of damage to dopaminergic neurons. Sustained ERS can activate apoptosis-related molecules such as CHOP and caspase 12, initiating the apoptosis program [62]. ERS and α-syn aggregates can also disrupt mitochondrial function, leading to a decrease in mitochondrial membrane potential, reduced ATP production, and increased ROS production. Mitochondrial dysfunction can further exacerbate ERS, forming a vicious cycle and jointly promoting the apoptosis of dopaminergic neurons [63]. ERS and α-syn aggregation further disrupt synaptic function by interfering with neurotransmitter release at the presynaptic membrane and receptor activity at the postsynaptic membrane (Figure 3). This leads to abnormal neurotransmission, impaired neuronal communication, and, ultimately, dopaminergic neuron dysfunction [64].

### 4.2. LRRK2 and ERS

The leucine-rich repeat kinase 2 (LRRK2) gene is one of the most common pathogenic genes for PD, and its mutations are associated with both familial and sporadic PD. The LRRK2 protein has multiple domains, including a protein kinase domain, a GTPase domain, a leucine-rich repeat domain, etc., and is involved in various intracellular signal transduction pathways and biological processes [65]. In recent years, studies have found that mutations in the LRRK2 gene can lead to ERS through interactions with thrombospondin-1 (THBS1)/transforming growth factor beta 1 (TGF-β1), thus playing an important role in the pathogenesis of PD [66].

Mutations in the LRRK2 gene lead to enhanced protein kinase activity, which is one of the key factors contributing to the pathogenesis of PD. The G2019S mutation is the most common mutation type in the LRRK2 gene, and this mutation significantly increases the protein kinase activity of LRRK2 [67]. Studies have shown that highly active LRRK2 can phosphorylate a variety of substrates, affecting intracellular signal transduction pathways [68]. Through the reprogramming of fibroblasts from PD patients with the LRRK2 G2019S mutation into induced pluripotent stem cells (iPSCs) and their differentiation into dopaminergic neurons, it was found that under the G2019S mutation condition, the sensitivity of dopaminergic neurons to ERS increased, manifested as the upregulation of the expression of ERS-related molecules such as GRP78 and CHOP, accompanied by impaired mitochondrial function [69]. After treatment with the LRRK2 inhibitor MLi-2, ERS and abnormal mitochondrial function can be relieved, indicating that the LRRK2 G2019S mutation is closely related to ERS [70,71].

Further studies have found that the LRRK2 G2019S mutation leads to ERS through interactions with THBS1/TGF-β1 [66]. Through big data analysis and functional experiments, the downstream targets regulated by LRRK2 G2019S were determined, and it was found that the expression of THBS1 increased significantly under mutation conditions [66]. Using the PYMOL technology, it was found that the LRRK2 protein can form a stable covalent bond with the THBS1 protein. THBS1 is a multifunctional glycoprotein that can interact with a variety of cell surface receptors and is involved in processes such as cell adhesion, migration, proliferation, and differentiation [72]. In PD, after LRRK2 binds to THBS1, it can activate the TGF-β1 signaling pathway [66]. The activation of the TGF-β1 signaling pathway affects the pathological process of the UPR, which is a hallmark of ERS, leading to the occurrence of ERS [73]. In a series of experimental verifications at the cellular level, knocking down THBS1 or inhibiting the TGF-β1 signaling pathway can alleviate the ERS induced by the LRRK2 G2019S mutation [74].

The research team also used the CRISPRi technology to knock out the THBS1 gene and the TGF-β1 gene in mice to construct mouse gene models, successfully replicating a series of results at the cellular level [66]. In mice with the LRRK2 G2019S mutation, after knocking out the THBS1 gene or the TGF-β1 gene, the ERS level of dopaminergic neurons in the mouse brain decreased, the damage to dopaminergic neurons was alleviated, and the motor function improved [66]. These results indicate that LRRK2 G2019S exerts a regulatory effect on ERS through THBS1/TGF-β1, affecting the disease progression of PD (Figure 3).

### 4.3. Other Related Molecules and ERS

In addition to α-syn and the LRRK2 gene, molecules such as Parkin and PTEN-induced kinase 1 (PINK1) also play important roles in PD-ERS [75,76] (Figure 3). Parkin is an E3 ubiquitin ligase encoded by the Parkin gene, and mutations in this gene are the main cause of autosomal recessive juvenile PD (AR-JP) [77]. The Parkin protein plays a key role in maintaining intracellular protein homeostasis and mitochondrial function [78]. In PD, there is a close connection between ERS and Parkin [79]. ERS can induce the upregulation of Parkin expression, which is an adaptive response of the cell. It was found that in the dopaminergic neuron cell model treated with MPP+, after ERS was activated, the expression of Parkin increased significantly [80]. The upregulated Parkin can degrade misfolded or unfolded proteins in the ER through the ubiquitin–proteasome system (UPS), reducing the burden on the ER and alleviating ERS [81]. Parkin can also be involved in mitophagy, clearing damaged mitochondria and reducing ROS production by mitochondria, thus reducing the damage of ERS to cells [82]. In the cell model with the Parkin gene knocked out, apoptosis induced by ERS increased significantly, indicating that Parkin plays an important protective role in maintaining ER homeostasis and cell survival [83].

PINK1 is a serine/threonine protein kinase encoded by the PINK1 gene [84]. Mutations in the PINK1 gene are also associated with the occurrence of PD [85]. PINK1 is mainly localized in mitochondria and plays an important role in maintaining mitochondrial function and homeostasis [86]. During the ERS process in PD, PINK1 is involved in regulating the interaction between mitochondria and the ER [83]. ERS can lead to mitochondrial Ca^2+^ overload, and PINK1 can maintain the balance of mitochondrial Ca^2+^ by regulating the Ca^2+^ transporters on the mitochondrial membrane, reducing the damage of ERS to mitochondria. PINK1 can also interact with Parkin to coordinately regulate mitophagy [87]. When mitochondria are damaged, PINK1 accumulates and is phosphorylated on the outer mitochondrial membrane, recruiting Parkin to the mitochondria and promoting the occurrence of mitophagy, thus clearing damaged mitochondria, reducing ROS production, and alleviating ERS [88]. Studies have shown that in cell models and animal models with PINK1 gene mutations, the sensitivity to ERS increases, mitochondrial dysfunction is aggravated, and the damage to dopaminergic neurons is exacerbated, further confirming the important role of PINK1 in PD-ERS [89].

**Figure 3 ijms-26-03367-f003:**
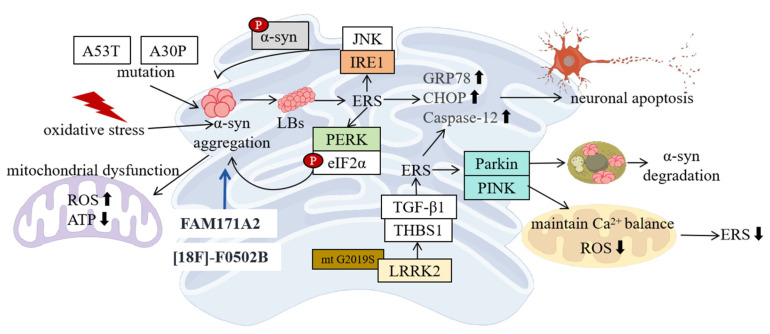
In PD, the molecular mechanism of ERS involves the interaction of multiple factors. The vicious cycle between alpha syn and ERS, where gene mutations or abnormal aggregation of alpha syn (such as A53T, A30P) lead to misfolding and the formation of LBs, directly triggering ERS (upregulation of GRP78 and CHOP expression). Meanwhile, ERS activates the UPR pathway, further promoting alpha syn aggregation and forming a positive feedback loop. Knocking out FAM171A2 can block the pathological spread of α-syn and improve symptoms. PET tracer [18F]-F0502B can visualize alpha syn aggregation. LRRK2 gene mutations drive ERS through the THBS1/TGF-β1 pathway. Common LRRK2 mutations (such as G2019S) enhance kinase activity, upregulate THBS1 expression, and bind to TGF-β1 to activate its signaling pathway, disrupting UPR homeostasis and leading to elevated ERS markers and mitochondrial damage. The synergistic protective effect of Parkin and PINK1, ERS-induced Parkin expression, clears misfolded proteins through the UPS and initiates mitochondrial autophagy, reducing the ERS burden; PINK1 regulates mitochondrial Ca^2+^ homeostasis and synergizes with Parkin to clear damaged mitochondria and reduce ROS production. Lack of Parkin or PINK1 can exacerbate ERS sensitivity and neuronal apoptosis. (“p” represents phosphorylation; ↑ represents increase; ↓ represents decrease).

## 5. Interaction Between ERS and Other Pathological Mechanisms of PD

### 5.1. Interaction with Mitochondrial Dysfunction

There is a close interaction between ERS and mitochondrial dysfunction in PD, which jointly promote the occurrence and development of the disease. From the perspective of the impact of ERS on mitochondrial function, ERS can disrupt mitochondrial function through multiple pathways [90]. When ERS occurs, the UPR is activated, which is, to a certain extent, an adaptive protective response of the cell. However, sustained ERS leads to the excessive activation of the UPR signaling pathway, thus affecting mitochondria [91]. ERS can lead to an imbalance in the Ca^2+^ transport between the ER and mitochondria. The ER is an important Ca^2+^ storage depot in the cell. Under normal circumstances, there is a close connection between the ER and mitochondria, and they communicate with each other through Ca^2+^ signals [92]. During ERS, the release of Ca^2+^ from the ER increases, resulting in mitochondrial Ca^2+^ overload. Mitochondrial Ca^2+^ overload can activate the mitochondrial permeability transition pore (mPTP), leading to a decrease in the mitochondrial membrane potential, reduced ATP production, increased ROS production, and further damage to mitochondrial function [93]. ERS can also damage the mitochondrial membrane by activating apoptosis-related signaling pathways, such as caspase 12, CHOP, etc., promoting the release of cytochrome C (cyt C) and triggering apoptosis [94]. In both cell models and animal models of PD, ERS-induced mitochondrial dysfunction has been observed. For example, in dopaminergic neurons treated with MPP+, after ERS is activated, the mitochondrial membrane potential decreases, ATP synthesis is reduced, the ROS level increases, and at the same time, there are changes in mitochondrial morphology, such as mitochondrial swelling and cristae breakage, etc. [95].

Mitochondrial dysfunction, in turn, exacerbates ERS. Mitochondria are the energy factories of the cell, and their dysfunction leads to abnormal energy metabolism and reduced ATP production [96]. Insufficient energy affects the normal function of the ER, as processes such as protein synthesis, folding, and transportation all require ATP to provide energy [97]. When mitochondrial dysfunction occurs, the production of ROS increases, and ROS can directly damage the ER, leading to the oxidative modification of ER proteins and affecting the structure and function of the ER [98]. In addition, mitochondrial dysfunction can also indirectly exacerbate ERS by activating the inflammatory response and apoptosis signaling pathways [99,100]. In the brain tissues of PD patients, mitochondrial dysfunction and ERS are often detected simultaneously, and both are related to the severity of the disease [101].

The ER–mitochondria contact sites (ERMCS) play a key role in this process. ERMCS are special structural regions between the ER and mitochondria. Through a series of protein interactions, they connect the ER and mitochondria closely [102]. ERMCS have important functions in regulating intracellular calcium homeostasis, lipid metabolism, mitochondrial dynamic balance, etc. [103]. In PD, the structure and function of ERMCS change abnormally. Studies have found that in the dopaminergic neurons of the substantia nigra of PD patients, the number of ERMCS increases and the structure is disordered, which may be the result of the interaction between ERS and mitochondrial dysfunction [104]. The abnormality of ERMCS leads to disruptions in communication and material exchange between the ER and mitochondria, further exacerbating ERS and mitochondrial dysfunction [105]. The ER cannot effectively transfer calcium ions to mitochondria, resulting in an imbalance of calcium homeostasis in mitochondria; mitochondria fail to provide feedback metabolites to the ER in a timely manner, affecting the normal function of the ER.

Some gene mutations related to PD also affect the interaction between the ER and mitochondria. For example, mutations in the PINK1 and Parkin genes are among the important causes of familial PD. PINK1 is a mitochondrial membrane protein, and its accumulation on the outer mitochondrial membrane can recruit Parkin to form a PINK1–Parkin complex, which, in turn, mediates the autophagic clearance of damaged mitochondria [106]. However, when mutations occur in the PINK1 or Parkin genes, the function of the PINK1–Parkin complex is impaired, and damaged mitochondria cannot be effectively cleared, leading to the exacerbation of mitochondrial dysfunction [106]. Mutations in the PINK1 and Parkin genes also affect the interaction between the ER and mitochondria, resulting in abnormal structure and function of ERMCS [107]. Studies have shown that in mice with the PINK1 or Parkin gene knocked out, the number of ERMCS decreases, and the calcium ion transfer and lipid exchange between the ER and mitochondria are blocked, further exacerbating ERS and mitochondrial dysfunction [108]. Similarly, mutations in the LRRK2 gene, such as the G2019S mutation, also disrupt the normal relationship between the ER and mitochondria. The LRRK2 G2019S mutation enhances kinase activity. It phosphorylates ERMCS tethering proteins such as VDAC1 and IP3R, which destabilizes the Ca^2+^ flux between the ER and mitochondria. This leads to ER Ca^2+^ depletion and subsequent ERS [109]. In summary, mutations in PD-associated genes, including LRRK2, PINK1, and Parkin, disrupt the structural and functional integrity of ERMCS. This exacerbates ER–mitochondrial miscommunication. For example, PINK1/Parkin loss-of-function mutations impair mitophagy, causing the accumulation of damaged mitochondria that overproduce ROS. ROS oxidize ER-resident proteins such as BiP, directly inducing ERS. Additionally, PINK1 deficiency reduces ERMCS density, disrupting lipid transfer and ER-to-mitochondria ATP shuttling, which aggravates ER protein misfolding [110,111].

The interconnection between ERS and mitochondrial dysfunction also involves the activation of the apoptosis signaling pathway. When ERS and mitochondrial dysfunction persist and cannot be effectively relieved, a series of apoptosis signaling pathways is activated, such as the caspase cascade reaction, the mitochondrial pathway, the ER pathway, etc., ultimately leading to the apoptosis of neurons [112]. The CHOP protein activated by ERS can promote the activation of proapoptotic proteins Bax and BAK by inhibiting the expression of the antiapoptotic protein Bcl-2, thus leading to the depolarization of the mitochondrial membrane potential, the release of cyt C, and the activation of caspase 9 and caspase 3 and triggering apoptosis [113]. The ROS generated by mitochondrial dysfunction also activate apoptosis signaling pathways such as JNK, further promoting the apoptosis of neurons.

The vicious cycle between ERS and mitochondrial dysfunction further exacerbates the damage to dopaminergic neurons. ERS leads to mitochondrial dysfunction, and mitochondrial dysfunction exacerbates ERS, worsening the intracellular environment and ultimately leading to the apoptosis of dopaminergic neurons [114]. Studies have shown that inhibiting ERS or improving mitochondrial function can reduce the damage to dopaminergic neurons, providing new targets for the treatment of PD (Figure 4).

### 5.2. Interaction with Oxidative Stress

ERS and oxidative stress promote each other and act synergistically in PD, jointly leading to neuronal damage. ERS is one of the important factors triggering oxidative stress [115]. When ERS occurs, unfolded proteins accumulate in large quantities in the ER lumen, activating the UPR signaling pathway. Under sustained ERS, the excessive activation of the UPR signaling pathway leads to the imbalance in the intracellular redox state, thus triggering oxidative stress [116]. ERS can lead to an increase in ROS production by activating the IRE1-JNK signaling pathway. After IRE1 is activated, it can phosphorylate and activate JNK, and JNK can activate NADPH oxidase, promoting the generation of ROS [117]. ERS can also lead to mitochondrial dysfunction. As mentioned above, mitochondrial dysfunction damages the mitochondrial respiratory chain and causes abnormal electron transfer, thus generating a large amount of ROS. In addition, during ERS, the function of the intracellular antioxidant defense system declines, and it fails to clear the excessive ROS in a timely manner, further exacerbating oxidative stress [82]. In cell models of PD, after overexpressing mutant α-syn or treating cells with ERS inducers, the intracellular ROS level increases significantly, and, at the same time, the expression of ERS-related molecules is upregulated [61].

Oxidative stress can also induce the occurrence of ERS. ROS are the main byproducts of oxidative stress and are highly chemically active. When the intracellular ROS level increases, it can directly damage the structure and function of the ER [118]. ROS can oxidatively modify the proteins and lipids in the ER, leading to the denaturation of ER proteins, loss of function, lipid peroxidation, and damage to the membrane structure of the ER [82]. Oxidative stress can also affect the homeostasis of Ca^2+^ in the ER. ROS can oxidatively modify the Ca^2+^ channel proteins in the ER, leading to Ca^2+^ leakage and a decrease in Ca^2+^ storage in the ER, thus triggering ERS [119]. In addition, oxidative stress can also activate intracellular stress signaling pathways, such as the p38 mitogen-activated protein kinase (p38 MAPK) pathway. The activation of these signaling pathways can further induce the expression of ERS-related molecules and exacerbate ERS [120]. In the brain tissues of PD patients, the markers of oxidative stress and ERS are significantly increased, indicating that the two coexist and interact with each other during the disease process.

ERS and oxidative stress promote each other and jointly damage dopaminergic neurons (Figure 4). The ROS generated by oxidative stress can damage biological macromolecules such as cell membranes, proteins, and nucleic acids, leading to the structural and functional impairment of dopaminergic neurons [121]. The apoptosis program triggered by ERS is also further activated by oxidative stress, promoting the apoptosis of dopaminergic neurons. The synergistic effect of the two also affects the synthesis, metabolism, and release of dopamine, resulting in abnormal dopaminergic neurotransmission and aggravating the symptoms of PD.

### 5.3. Interaction with Neuroinflammation

ERS and neuroinflammation are interrelated and influence each other in PD, jointly participating in the pathological process of the disease. ERS can trigger neuroinflammation. In PD, after ERS is activated, it can induce neuroinflammatory responses through multiple pathways [122]. ERS can lead to the accumulation of misfolded proteins in the cell, and these misfolded proteins can be recognized by the cell as pathogen-associated molecular patterns (PAMPs) or damage-associated molecular patterns (DAMPs), thus activating the innate immune response [123]. The cell then releases some danger signal molecules, such as high-mobility group box 1 protein (HMGB1). These molecules can bind to the pattern recognition receptors (PRRs) on the surface of immune cells, such as toll-like receptors (TLRs), and then activate the immune cells, triggering neuroinflammation [124]. ERS can also activate the nuclear factor kappa-B (NF-κB) signaling pathway. Under normal circumstances, NF-κB binds to its inhibitory protein IκB and exists in the cytoplasm in an inactive form [125]. During ERS, the IRE1 pathway is activated, and through a series of signal transduction processes, the IκB kinase (IKK) is activated. IKK phosphorylates IκB, causing its degradation, thus releasing NF-κB [125]. After NF-κB enters the nucleus, it can activate the transcription of a series of inflammation-related genes, such as the expression of inflammatory factors such as tumor necrosis factor-α (TNF-α), interleukin-1β (IL-1β), IL-6, etc., leading to neuroinflammation. In both animal models of PD and the brain tissues of patients, a close association between ERS and neuroinflammation has been observed [126]. For example, in the MPTP-induced PD mouse model, while the expression of ERS-related molecules in the mouse brain is upregulated, the levels of neuroinflammatory factors also increase significantly [126].

Neuroinflammation, in turn, exacerbates ERS (Figure 4). During the neuroinflammatory process, activated microglia and astrocytes release a large amount of inflammatory factors, such as TNF-α, IL-1β, IL-6, etc. [127,128]. These inflammatory factors can act on neurons and glial cells, affecting the normal function of the ER and exacerbating ERS. TNF-α can inhibit the folding and transportation of proteins in the ER, leading to the accumulation of unfolded proteins and thus exacerbating ERS [129]. IL-1β can activate the intracellular stress signaling pathways, such as the JNK and p38 MAPK pathways. The activation of these signaling pathways can induce the expression of ERS-related molecules and further exacerbate ERS [130]. Neuroinflammation can also lead to an increase in oxidative stress. As mentioned above, oxidative stress can damage the ER, thus indirectly exacerbating ERS. In the brains of PD patients, neuroinflammation and ERS promote each other, forming a vicious cycle and jointly promoting the damage to dopaminergic neurons and the progression of the disease.

**Figure 4 ijms-26-03367-f004:**
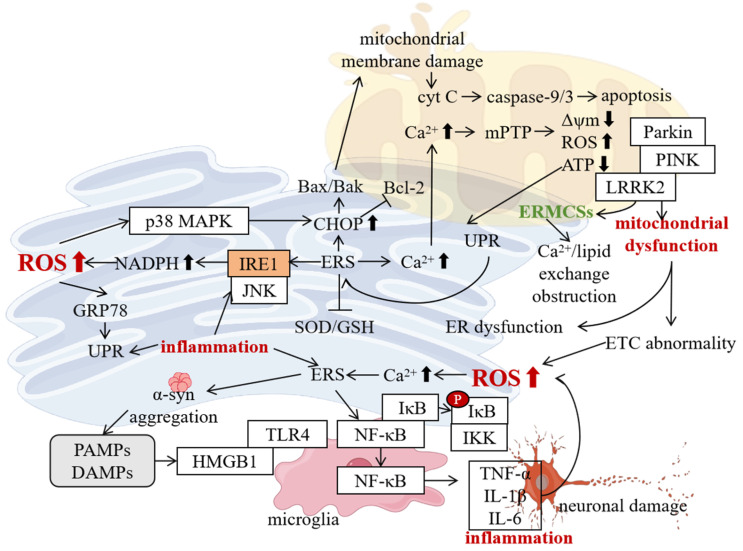
In PD, ERS forms a complex vicious cycle with mitochondrial dysfunction, oxidative stress, and neuroinflammation, collectively driving disease progression. ERS activates the mPTP via calcium overload, leading to mitochondrial membrane potential collapse, reduced ATP synthesis, and a ROS burst, while simultaneously activating proapoptotic pathways (CHOP/Bax/Bak) to exacerbate mitochondrial damage. Mitochondrial dysfunction-induced energy metabolism collapse and ROS feedback further disrupt ER homeostasis, creating a bidirectional detrimental loop. Structural disorganization of ERMCS and mitophagy defects caused by PINK1/Parkin mutations amplify this cross-damage. Additionally, ERS activates NADPH oxidase through the IRE1-JNK pathway, while mitochondrial ROS leakage directly triggers oxidative stress. ROS oxidizes ER proteins and calcium channels, impairing their function and forming a mutually reinforcing cycle, compounded by weakened antioxidant systems (SOD/GSH). ERS also activates the TLR/NF-κB pathway via misfolded protein release (e.g., α-syn), inducing pro-inflammatory cytokines (TNF-α, IL-1β), which, in turn, suppress ER folding capacity and activate JNK/p38 pathways to aggravate ERS. Microglial activation synergizes with oxidative stress, forming an “ERS–inflammation–oxidative stress” triangular vicious cycle. Ultimately, apoptotic cascades, metabolic imbalance, and synaptic dysfunction lead to irreversible dopaminergic neuron loss. Targeting key nodes in these interconnected networks (e.g., inhibiting ERS, enhancing mitophagy, or blocking inflammatory signaling) may offer novel therapeutic strategies for PD. (“p” represents phosphorylation; ↑ represents increase; ↓ represents decrease).

## 6. Exploration of PD Treatment Strategies Based on ERS

### 6.1. Drug Treatment Strategies

The important role of ERS in PD is not only reflected in its pathological mechanism, but also provides new ideas for its diagnosis and treatment. ERS markers, such as GRP78, CHOP, and XBP-1, are significantly upregulated in the brains of PD patients. The detection of these markers may serve as a biomarker for the early diagnosis of PD [131]. By detecting the expression levels of these markers, PD patients can be identified earlier, enabling early intervention and treatment. In terms of treatment, intervention strategies targeting ERS show potential clinical application value. For example, the use of chemical chaperones such as 4-phenylbutyric acid (4-PBA) and tauroursodeoxycholic acid (TUDCA) can alleviate ERS, improve protein folding and transportation, and thus protect dopaminergic neurons [132]. In addition, inhibiting the apoptosis signaling pathways mediated by the UPR, such as inhibitors of CHOP and caspase 12, may also offer new methods for treating PD [133].

Some drugs act by regulating the IRE1 pathway. For example, 4μ008C is an IRE1 inhibitor that can specifically inhibit the ribonuclease activity of IRE1, thereby blocking the IRE1-XBP-1 pathway [134]. In cell models and animal models of PD, the application of 4μ8C can reduce the splicing of XBP-1, lower the expression of ERS-related proteins, and alleviate the damage to dopaminergic neurons [134]. This indicates that 4μ8C may relieve ERS by inhibiting the IRE1 pathway and has potential as a therapeutic agent for PD.

Drugs targeting the PERK pathway are also under study. GSK2606414 is a PERK inhibitor that can inhibit the activity of PERK and reduce the phosphorylation of eIF2α, thereby restoring protein synthesis [135]. In the PD model, treatment with GSK2606414 can reduce the level of ERS, reduce the apoptosis of dopaminergic neurons, and improve the motor function of animals. These results suggest that GSK2606414 could be a potential drug for treating PD. Natural products also show promise in the treatment of PD. The bidirectional relationship between mitochondrial dysfunction and ERS necessitates combination therapeutic strategies. For example, the PERK inhibitor GSK2606414, while alleviating ERS, must be paired with mitochondrial antioxidants (e.g., coenzyme Q10) to address the root causes of stress [136]. Similarly, enhancing mitophagy via PINK1/Parkin activators (e.g., kinetin) could reduce mitochondrial ROS-driven ERS, creating a synergistic effect with chemical chaperones such as tauroursodeoxycholate (TUDCA) [137]. Future clinical trials should prioritize dual-target approaches to disrupt the self-perpetuating cycle of organelle stress in PD.

Many natural products have multiple biological activities, such as antioxidant, anti-inflammatory, and ERS-regulating properties. Oxymatrine is a compound derived from the dried roots of *Sophora flavescens*, a plant of the Leguminosae family, and has been used in traditional Chinese medicine for hundreds of years. Studies have found that oxymatrine, combined with a soluble epoxide hydrolase (sEH) inhibitor, can reduce neuroinflammation in animal models of PD [138]. Further studies have shown that oxymatrine may play a role in treating PD by regulating ERS and reducing the damage to dopaminergic neurons. Another promising compound is withaferin A, a natural pharmaceutical compound with significant antioxidant and anti-inflammatory functions. Studies have shown that for traditional Parkinson’s model mice and humanized Parkinson’s model mice, withaferin A can support the synthesis of dopamine in the substantia nigra neurons of the midbrain and slow down the PD-like damage to these neurons. Its mechanism of action may be related to relieving ERS and reducing the apoptosis of dopaminergic neurons [139].

### 6.2. Non-Drug Treatment Strategies

In addition to drug treatment, non-drug treatment methods also play an important role in the treatment of PD. Electroacupuncture, as a traditional Chinese medical therapy, has been increasingly studied in the treatment of PD in recent years, and it has been found that it can regulate ERS in PD [140].

Relevant experiments have studied the effect of electroacupuncture at the “Fengfu” and “Taichong” acupoints on ERS-related proteins in PD model rats. By establishing PD model rats and performing electroacupuncture interventions on them, collecting brain tissue samples from the rats, and using methods such as immunohistochemistry, Western blotting, and real-time fluorescence quantitative PCR for detection, it has been found that electroacupuncture at the “Fengfu” and “Taichong” acupoints can significantly improve the behavioral performance of PD rats, reduce pathological damage, and decrease the expression levels of ERS-related proteins [140]. This indicates that electroacupuncture may protect neurons from damage and death by reducing ERS and decreasing the accumulation of ruptured and aggregated proteins. The study also set up different electroacupuncture treatment duration groups, and the results showed that with the extension of the treatment time, the therapeutic effect of electroacupuncture became more obvious. Compared with the electroacupuncture treatment group treated for 7 days, the electroacupuncture treatment group treated for 28 days had a higher positive expression of TH in the SN area of rats and a lower expression of ERS-related proteins ATF6 and CHOP. This shows that electroacupuncture treatment for PD has a certain time-dependent effect, and a longer treatment time may be more conducive to regulating ERS and improving the symptoms of PD [140].

Although the specific mechanism by which electroacupuncture regulates ERS has not been fully clarified at present, it is believed that it may be related to acupuncture regulating neurotransmitters, improving energy metabolism, and activating protective signaling pathways within cells. Electroacupuncture stimulation of acupoints may transmit signals through the nervous system and regulate the release of neurotransmitters in the brain, such as dopamine and γ-aminobutyric acid. The balance of these neurotransmitters is crucial for maintaining the normal function of neurons. Electroacupuncture may also improve the energy metabolism of cells and reduce the stress on the ER caused by insufficient energy. In addition, electroacupuncture may activate the protective signaling pathways within cells, such as the phosphatidylinositol 3-kinase (PI3K)–protein kinase B (AKT) pathway, enhancing the cells’ resistance to stress and thus reducing ERS.

ERS is also closely related to other pathological features of PD, such as mitochondrial dysfunction and neuroinflammation. Therefore, treatment strategies targeting these pathological processes may also indirectly reduce ERS. For example, antioxidants and anti-inflammatory drugs can alleviate mitochondrial dysfunction and neuroinflammation, thereby relieving ERS-induced damage to dopaminergic neurons [140,141]. In conclusion, ERS has significant potential for application in the diagnosis and treatment of PD. By detecting ERS markers and developing treatment strategies targeting ERS, we hope to provide more effective diagnostic and treatment methods for PD patients.

## 7. Conclusions

ERS plays a central role in the pathological process of PD, especially in the loss of dopaminergic neurons and the abnormal aggregation of α-syn. Through the UPR, the cell attempts to restore ER homeostasis, but if the stress persists, the cell initiates the apoptosis program, leading to neuronal death. ERS is also closely related to other pathological features of PD, such as mitochondrial dysfunction and neuroinflammation, which further exacerbate neuronal damage. The molecular mechanisms of ERS involve the activation of signaling pathways such as IRE1, PERK, and ATF6, as well as the expression of apoptosis-related proteins such as CHOP and caspase 12. These mechanisms not only reveal the important role of ERS in PD, but also provide new ideas for diagnosis and treatment. The detection of ERS markers is expected to become a biomarker for the early diagnosis of PD, and intervention strategies targeting ERS, such as chemical chaperones and inhibitors of the apoptosis signaling pathway, also show potential for clinical application. Future research needs to further explore the specific mechanisms of ERS in PD and develop more effective diagnostic and treatment methods. The latest cutting-edge research achievements have brought us new hope and new directions. The therapeutic target FAM171A2, newly discovered by the team of Yu Jintai from Huashan Hospital Affiliated to Fudan University, and the PD PET molecular imaging biomarker developed by Renmin Hospital of Wuhan University have shown high potential value in the early diagnosis and treatment intervention of PD. Innovative treatment ideas based on ERS, such as regulating the UPR signaling pathway, reducing the accumulation of misfolded proteins, and improving the ER–mitochondrial function, also provide new strategies and methods for the treatment of PD.

In the future, research on ERS and its molecular mechanisms in PD could be carried out in the following directions. In terms of in-depth mechanistic research, although certain achievements have been made, there are still many unknown areas to be explored. Further research on the fine regulatory mechanisms of the ERS signaling pathway and the complex interactions between ERS and other pathological mechanisms will help develop a deeper understanding of the pathogenesis of PD. Studying the changing patterns of ERS at different stages of PD and how to intervene in ERS at an early stage to delay disease progression are also important research directions for the future. In terms of the research and development of new therapeutic targets and drugs, continuing to search for new therapeutic targets related to ERS and developing more effective drugs are the key research points for the future. It is also important to combine new technologies such as artificial intelligence and big data to accelerate the drug research and development process and improve research and development efficiency, as well as explore combination treatment strategies, combining treatments targeting ERS with other treatment methods to achieve better treatment effects. In terms of clinical translational research, translating basic research achievements into clinical treatment methods is the ultimate goal, as well as conducting clinical trials to verify the safety and effectiveness of ERS-based treatment strategies and providing more effective treatment methods for PD patients and establishing ERS-related biomarkers for PD for the early diagnosis, disease monitoring, and prognosis assessment of the disease to improve the clinical diagnosis and treatment level. The research on ERS and its molecular mechanisms in PD has important theoretical and clinical significance. Through in-depth research, it is expected to bring new breakthroughs in the treatment of PD and improve the quality of life of patients.

## Figures and Tables

**Figure 1 ijms-26-03367-f001:**
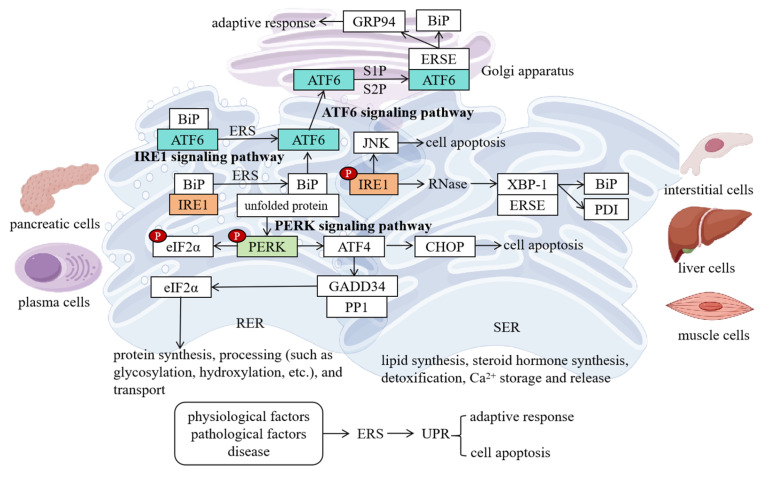
The ER is a membranous network structure in eukaryotic cells, divided into the RER and the SER. The RER, studded with ribosomes, is responsible for the synthesis and modification of secretory proteins (e.g., glycosylation) and is highly developed in cells with active secretory functions. The SER lacks ribosomes and performs diverse roles, including steroid hormone synthesis (e.g., in adrenal cells), detoxification (e.g., in liver cells), calcium ion storage (e.g., in the sarcoplasmic reticulum regulating muscle contraction), and membrane lipid synthesis. The ER also participates in cellular signaling through calcium homeostasis regulation. When ER homeostasis is disrupted (e.g., due to hypoxia, misfolded protein accumulation, or calcium imbalance), ERS is triggered, prompting the cell to activate the UPR via three core pathways: (1) the IRE1 pathway, upon activation, cleaves XBP-1 mRNA to generate the transcription factor XBP-1, which upregulates chaperones (e.g., BiP) to enhance the protein-folding capacity; prolonged stress activates the JNK pathway to promote apoptosis; (2) the PERK pathway phosphorylates eIF2α to inhibit global protein synthesis while inducing ATF4 expression to activate stress adaptation genes (e.g., antioxidant genes); chronic activation upregulates the pro-apoptotic factor CHOP; (3) the ATF6 pathway is transported to the Golgi apparatus, where it is cleaved to release its active fragment, which enters the nucleus to activate ER function-related genes. These pathways synergistically reduce the unfolded protein burden and restore homeostasis. If stress persists and remains unresolved, the UPR shifts toward apoptosis. (“p” represents phosphorylation).

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
