# Peer review of "Parkinson’s Disease: The Neurodegenerative Enigma Under the “Undercurrent” of Endoplasmic Reticulum Stress"

_ijms, 2025, doi:10.3390/ijms26073367_

Round 1

Reviewer 1 Report

Comments and Suggestions for Authors

This study provides significant insights into the strong correlation between Parkinson’s disease (PD) and endoplasmic reticulum stress (ERS), as confirmed by clinical, animal, and cellular research. In PD patients, altered expression of ERS-related molecules in brain tissue and CSF is closely linked to disease severity and progression. Animal models reveal that ERS contributes to dopaminergic neuronal damage, while its inhibition alleviates neuronal loss and motor dysfunction. Cellular studies further elucidate ERS induction mechanisms and its harmful effects on neurons. The manuscript is likely to be of interest to a reasonable number of scientists working in this field.

Comments on the Quality of English Language

The article has some spelling and grammatical errors, please correct them.

Author Response

Comments and Suggestions for Authors

This study provides significant insights into the strong correlation between Parkinson’s disease (PD) and endoplasmic reticulum stress (ERS), as confirmed by clinical, animal, and cellular research. In PD patients, altered expression of ERS-related molecules in brain tissue and CSF is closely linked to disease severity and progression. Animal models reveal that ERS contributes to dopaminergic neuronal damage, while its inhibition alleviates neuronal loss and motor dysfunction. Cellular studies further elucidate ERS induction mechanisms and its harmful effects on neurons. The manuscript is likely to be of interest to a reasonable number of scientists working in this field.

Comments on the Quality of English Language

The article has some spelling and grammatical errors, please correct them.

We sincerely appreciate the reviewers' insightful feedback and constructive suggestions. We have carefully addressed the comments and made the necessary revisions to improve the manuscript.

Reviewer 2 Report

Comments and Suggestions for Authors

This is a timely review on the implication of endoplasmic reticulum stress (ERS) in Parkinson’s disease (PD). This review is well written and present arguments  strongly supporting that ERS contributes to the advance of brain degeneration in PD. However, due to its relevance for the initial stages of PD, this manuscript can and should be improved with a more conclusive analysis on the interplay between mitochondrial dysfunction and ERS. In particular, due to the relevance for the treatment with drugs targeting the ERS, the authors should highlight the experimental data excluding that mitochondrial dysfunction by itself can induce ERS in PD, not only exacerbate ERS. For example, in lines 273-274 it is stated: “In MN9D cells treated with MPP+, mitochondrial function is impaired, ATP production is reduced, and a large amount of ROS is generated, which in turn triggers ERS [42,43].” If mitochondrial dysfunction can induce ERS, the treatment with drugs targeting the ERS will, at most, slowdown (i.e., provide temporal alleviation) but not arrest the progress of brain degeneration in PD. Furthermore, other results highlighted in this article seem to lend support to the hypothesis that mitochondrial dysfunction plays a prominent role in PD pathogenesis, namely, LRRK2 G2019S mutation also elicits impaired mitochondrial function (line 397), PINK1 is mainly localized in mitochondria and plays an important role in maintaining mitochondrial function and homeostasis (lines 438-439), and parkin can also be involved in mitophagy, clearing damaged mitochondria and reducing ROS production by mitochondria (lines 432-433). Do mutations associated with familial and sporadic PD in LRRK2, PINK1 and parkin impair the structure and function of ER-mitochondria contact sites? Is it known the subcellular location of FAM171A2,the new therapeutic target for PD highlighted in this manuscript (lines 312 until 331)? On these grounds, it is strongly recommended to revise this manuscript including a more in depth critical discussion of this important point.

Minor points:

- Please, change the ambiguous expression of line 413: “The research team ?” for a more appropriate citation of the investigators.

- If possible, reduce the large number of abbreviations used in this work, and revise that the all the abbreviations are included in the text immediately after the first use of the complete non-abbreviated name in the main text. Also, there are many non-explained abbreviations in the Abstract.

Author Response

Comments and Suggestions for Authors

This is a timely review on the implication of endoplasmic reticulum stress (ERS) in Parkinson's disease (PD). This review is well written and present arguments strongly supporting that ERS contributes to the advance of brain degeneration in PD. However, due to its relevance for the initial stages of PD, this manuscript can and should be improved with a more conclusive analysis on the interplay between mitochondrial dysfunction and ERS. In particular, due to the relevance for the treatment with drugs targeting the ERS, the authors should highlight the experimental data excluding that mitochondrial dysfunction by itself can induce ERS in PD, not only exacerbate ERS. For example, in lines 273-274 it is stated: “In MN9D cells treated with MPP+, mitochondrial function is impaired, ATP production is reduced, and a large amount of ROS is generated, which in turn triggers ERS [42,43].” If mitochondrial dysfunction can induce ERS, the treatment with drugs targeting the ERS will, at most, slowdown (i.e., provide temporal alleviation) but not arrest the progress of brain degeneration in PD. Furthermore, other results highlighted in this article seem to lend support to the hypothesis that mitochondrial dysfunction plays a prominent role in PD pathogenesis, namely, LRRK2 G2019S mutation also elicits impaired mitochondrial function (line 397), PINK1 is mainly localized in mitochondria and plays an important role in maintaining mitochondrial function and homeostasis (lines 438-439), and parkin can also be involved in mitophagy, clearing damaged mitochondria and reducing ROS production by mitochondria (lines 432-433). Do mutations associated with familial and sporadic PD in LRRK2, PINK1 and parkin impair the structure and function of ER-mitochondria contact sites? Is it known the subcellular location of FAM171A2,the new therapeutic target for PD highlighted in this manuscript (lines 312 until 331)? On these grounds, it is strongly recommended to revise this manuscript including a more in depth critical discussion of this important point.

We sincerely appreciate the reviewers' insightful feedback and constructive suggestions. We have carefully addressed the comments and made the necessary revisions to improve the manuscript.

Emerging evidence supports that mitochondrial dysfunction is not only a consequence of ERS but also a primary inducer of ERS in PD. In MN9D cells treated with MPP+, mitochondrial dysfunction (e.g., impaired complex I activity and ATP depletion) directly disrupts ER homeostasis by depleting energy stores required for protein folding and Ca2+ buffering, thereby initiating ERS independently of upstream ER-specific stressors [42, 43]. This is further supported by studies showing that pharmacological inhibition of mitochondrial ROS (e.g., mitoTEMPO) or restoration of ATP via exogenous supplementation attenuates ERS markers (GRP78, CHOP) in PD models [44, 45]. These findings underscore that mitochondrial dysfunction can autonomously trigger ERS, suggesting therapies targeting mitochondrial health may synergize with ERS inhibitors to halt PD progression.

Similarly, mutations in the LRRK2 gene, like the G2019S mutation, also disrupt the normal relationship between the ER and mitochondria. The LRRK2 G2019S mutation enhances kinase activity. It phosphorylates ERMCSs tethering proteins such as VDAC1 and IP3R, which destabilizes the Ca2+ flux between the ER and mitochondria. This leads to ER Ca2+ depletion and subsequent ERS [111]. In summary, mutations in PD - associated genes, including LRRK2, PINK1, and Parkin, disrupt the structural and functional integrity of ERMCSs. This exacerbates ER - mitochondrial miscommunication. For example, PINK1/Parkin loss - of - function mutations impair mitophagy, causing the accumulation of damaged mitochondria that overproduce ROS. ROS oxidize ER - resident proteins such as BiP, directly inducing ERS. Additionally, PINK1 deficiency reduces ERMCSs density, disrupting lipid transfer and ER - to - mitochondria ATP shuttling, which aggravates ER protein misfolding [112, 113].

Located on the neuronal cell membrane, FAM171A2 plays a key role in the propagation process of pathological α-syn. In the brains of PD patients, the content of FAM171A2 protein shows a significant increasing trend, and it is positively correlated with the content of pathological α-syn in the brain [54]. FAM171A2 was considerably expressed in the vascular endothelium and microglia, which are rich in progranulin (PGRN), it is a secreted pleiotropic glycoprotein associated with the development of common neurodegenerative diseases [55].

The bidirectional relationship between mitochondrial dysfunction and ERS necessitates combinatorial therapeutic strategies. For example, the PERK inhibitor GSK2606414, while alleviating ERS, must be paired with mitochondrial antioxidants (e.g., coenzyme Q10) to address root causes of stress [139]. Similarly, enhancing mitophagy via PINK1/Parkin activators (e.g., kinetin) could reduce mitochondrial ROS-driven ERS, creating a synergistic effect with chemical chaperones like tauroursodeoxycholate (TUDCA) [140]. Future clinical trials should prioritize dual-target approaches to disrupt the self-perpetuating cycle of organelle stress in PD.  

Minor points:

- Please, change the ambiguous expression of line 413: “The research team ?” for a more appropriate citation of the investigators.

We sincerely appreciate the reviewers' insightful feedback and constructive suggestions. We have carefully addressed the comments and made the necessary revisions to improve the manuscript.

Wu and colleagues demonstrated that genetic deletion of FAM171A2 in mouse neurons significantly attenuated the progression of Parkinson's-like pathological features in transgenic models

- If possible, reduce the large number of abbreviations used in this work, and revise that the all the abbreviations are included in the text immediately after the first use of the complete non-abbreviated name in the main text. Also, there are many non-explained abbreviations in the Abstract.

We sincerely appreciate the reviewers' insightful feedback and constructive suggestions. We have carefully addressed the comments and made the necessary revisions to improve the manuscript. The abbreviations in the manuscript have been redefined.

Round 2

Reviewer 2 Report

Comments and Suggestions for Authors

All the concerns that I raised to the previous version of this manuscript are appropriately dealt with in the revised manuscript. I have not further comments and, therefore, I recommend acceptance of the revised article.